# ^68^Ga-PSMA-11 PET/CT-Guided Stereotactic Body Radiation Therapy Retreatment in Prostate Cancer Patients with PSA Failure after Salvage Radiotherapy

**DOI:** 10.3390/biomedicines8120536

**Published:** 2020-11-25

**Authors:** Paola Caroli, Sarah Pia Colangione, Ugo De Giorgi, Giulia Ghigi, Monica Celli, Emanuela Scarpi, Manuela Monti, Valentina Di Iorio, Anna Sarnelli, Giovanni Paganelli, Federica Matteucci, Antonino Romeo

**Affiliations:** 1Unit of Nuclear Medicine, Istituto Scientifico Romagnolo per lo Studio e la cura dei Tumori (IRST) IRCCS, 47014 Meldola, Italy; paola.caroli@irst.emr.it (P.C.); monica.celli@irst.emr.it (M.C.); giovanni.paganelli@irst.emr.it (G.P.); 2Department of Radiotherapy, Istituto Scientifico Romagnolo per lo Studio e la cura dei Tumori (IRST) IRCCS, 47014 Meldola, Italy; sarah.colangione@auslromagna.it (S.P.C.); giulia.ghigi@irst.emr.it (G.G.); antonino.romeo@irst.emr.it (A.R.); 3Department of Medical Oncology, Istituto Scientifico Romagnolo per lo Studio e la cura dei Tumori (IRST) IRCCS, 47014 Meldola, Italy; ugo.degiorgi@irst.emr.it; 4Unit of Biostatistics and Clinical Trials, Istituto Scientifico Romagnolo per lo Studio e la cura dei Tumori (IRST) IRCCS, Via Maroncelli 40, 47014 Meldola, Italy; emanuela.scarpi@irst.emr.it (E.S.); manuela.monti@irst.emr.it (M.M.); 5Oncology Pharmacy, Istituto Scientifico Romagnolo per lo Studio e la cura dei Tumori (IRST) IRCCS, 47014 Meldola, Italy; valentina.diiorio@irst.emr.it; 6Unit of Medical Physics, Istituto Scientifico Romagnolo per lo Studio e la cura dei Tumori (IRST) IRCCS, 47014 Meldola, Italy; anna.sarnelli@irst.emr.it

**Keywords:** 68Ga-PSMA PET/CT, prostate cancer, radiotherapy, re-treatment

## Abstract

(1) Purpose: To investigate the role of ^68^Ga-PSMA-11 PET/CT in guiding retreatment stereotactic body radiation therapy (SBRT) in prostate cancer (PCa) patients in biochemical recurrence (BCR) after salvage radiotherapy (S-RT). (2) Methods: We retrospectively evaluated PCa patients previously treated with S-RT on the prostate bed and with proven serum prostate antigen (PSA) failure after S-RT. In all patients (pts), ^68^Ga-PSMA-11 PET/CT was positive in the prostate bed only and guided retreatment SBRT. All retreatments were performed by applying the same radiotherapy protocol (median dose of 18 Gy/3 fractions; IQR 18–21 Gy). The median follow-up was 27 months (range 4–35 months). (3) Results: 38 consecutive patients were considered in this analysis. The overall median PSA level before RT was 1.10 ng/mL (IQR 0.82–2.59). PSA decreased at 3 and 6 months after treatment, with a median value of 0.60 ng/mL (IQR 0.31–0.96; *p* < 0.001) and 0.51 ng/mL (IQR 0.29–1.17; *p* < 0.001), respectively. Overall, biochemical recurrence-free survival (b-RFS) was 15.0 months (95% CI 13–23). Grade-1 toxicity was reported in 31.6% of patients (12/38). (4) Conclusion: These results confirm that ^68^Ga-PSMA-11-PET/CT is able to identify the site of recurrence in patients who have failed S-RT, thus supporting the use of metastases-directed radiotherapy as a safe and effective treatment.

## 1. Introduction

Prostate cancer (PCa) is the most frequent malignancy in male patients [1]. Despite radical approaches such as prostatectomy and radiation therapy, 20–30% of PCa patients will develop a biochemical relapse (BCR) within 10 years from primary care, defined by progressively rising serum prostate antigen (PSA) levels [2]. Salvage radiation therapy to the prostatic bed and/or pelvic nodes can offer a chance for disease control at first BCR. However, many patients treated with standard salvage radiation therapy experience PSA failure as a result of new recurrences of active disease; the mainstay for disease control in such patients is systemic therapy based on hormone blockade. Despite its efficacy, hormone-based therapy often entails poorly tolerated symptoms in BCR patients and can facilitate significant comorbidity, leading to therapy interruptions and/or cardiovascular and metabolic complications [3]. In early BCR scenarios, namely, for serum PSA levels as low as 1.0 ng/mL, patients may harbor low burden disease within the pelvis and/or at a few extrapelvic sites, deemed to be treatable with local therapy when the site of recurrence is localized [4].

Recent advances in imaging techniques have introduced a new chance for disease control and/or systemic therapy procrastination. In particular, the introduction in clinical practice of prostate-specific membrane antigen-based positron emission computed tomography hybrid imaging (^68^Ga-PSMA-11 PET/CT) has demonstrated high sensitivity and specificity for disease recurrence localization in patients with a low burden of the disease. ^68^Ga-PSMA-11 PET/CT holds higher diagnostic accuracy compared to conventional imaging (bone scan and CT imaging), choline PET/CT, and fluciclovine PET/CT [5,6].

At present, technological evolution and refinement of radiation therapy techniques allow us to perform imaging-guided retreatments in previously irradiated disease sites with good efficacy and toxicity profiles [7]. Thus, the integration of ^68^Ga-PSMA-11 PET/CT imaging to guided radiation therapy planning offers the possibility of identifying and treating site(s) of clinical relapse in patients with low-PSA biochemical relapse, who already failed salvage radiotherapy (S-RT) in the prostate bed.

The aim of this study is to investigate the role of ^68^Ga-PSMA-11 PET/CT in guiding retreatment (SBRT) in PCa patients affected by PSA failure after S-RT.

## 2. Materials and Methods

### 2.1. Patient Population

As part of the IRST 182.04 prospective study between June 2016 and December 2018, we evaluated 76 consecutive postsalvage BCR patients sent by the radiotherapy unit to perform ^68^Ga-PSMA-11 PET/CT with the aim of identifying disease recurrence amenable to local radiation treatment.

All procedures performed in studies involving human participants were in accordance with the ethical standards of the institutional and/or national research committee and the principles of the 1964 Declaration of Helsinki and its later amendments or comparable ethical standards. Written informed consent was obtained from all participants at enrollment, along with signed participant assent, when applicable. Consent to publish was obtained from all participants or guardians at enrollment along with signed participant assent, when applicable, as part of the informed consent form. This study was approved by IRST-Area Vasta Romagna ethics committee 2016 June 15 (protocol IRST 182.04 Eudract number 2015-003397-33).

The patients had previously undergone either curative prostatic radiotherapy or radical prostatectomy with pelvic lymphadenectomy, followed by adjuvant/salvage radiotherapy for prostate cancer.

All patients were M0 on initial histology and had not been on hormone therapy for at least six months.

Inclusion criteria were (a) previous S-RT on the prostate bed; (b) proven PSA failure after S-RT; (c) ^68^Ga-PSMA-11 PET/CT performed after S-RT was positive in the prostate bed only; (d) radiotherapy retreatment guided by ^68^Ga-PSMA-11 PET/CT; (e) complete clinical follow-up available.

From the present analysis, we excluded patients who were negative on ^68^Ga-PSMA-11 PET/CT (16/76 patients, 21%) and patients with ^68^Ga-PSMA-11 PET/CT positivity outside the locoregional bed (17/76 patients, 22%), namely, 13 patients with retroperitoneal nodal recurrence and 4 patients with bone metastases.

Based on the grading of the surgical histological examination, we divided the patients into three different subsets: Subset 1, which includes patients with ISUP 1 and 2 (*n* = 23); Subset 2, patients with ISUP 3 (*n* = 15); Subset 3 (*n* = 5), with ISUP 4 and 5.

The 5 patients belonging to Subset 3 were excluded from analysis as androgen deprivation therapy (ADT) had been administered in accordance with international procedural guidelines. 

The final population comprised 38 patients, with ^68^Ga-PSMA-11 PET/CT evidence of locoregional recurrence. 

All patients were monitored with repeat serum PSA measurements every 3 months after radiation retreatment and clinical follow-up visits until evidence of new PSA arose. All patients studied were followed-up, with a median time of 27 months (range 4–35 months).

### 2.2. Preparation of ^68^Ga- PSMA-11

^68^Ga-HBED-CC-PSMA was prepared according to national regulations, good radiopharmaceutical practices (GRP), as defined by EANM guidelines, and “Good Manufacturing Procedure (GMP) in Nuclear Medicine” [8].

^68^Ga-PSMA-11 was synthesized with the Modular-Lab Eazy kit (Eckert & Ziegler Radiopharma GmbH, Berlin, Germany) without organic solvents and with sterile disposable cassettes. Radionuclide ^68^Gallium (^68^Ga) was routinely obtained as Gallium chloride (^68^GaCl_3_) solution by elution of a commercial ^68^Ge/^68^Ga generator, 1.11 GBq (GalliaPharm^®^, Eckert & Ziegler Radiopharma GmbH, Berlin, Germany). PSMA-11 was purchased from Advanced Biochemical Compounds, ABX (Radeberg, Germany).

The first step of radiolabeling is the elution of ^68^GaCl_3_ from the generator. Then ^68^Ga is trapped in a prepurification SCX cartridge with strong cation exchange resin in order to minimize the presence of metal impurities from the ^68^GaCl3 eluted, while HCl is directed to the waste vial.

Subsequently, ^68^Ga is eluted with NaCl solution 5 M, acidified with HCl and transferred to the reaction vial that contains the reaction mixture: 400 µl of acetate buffer pH 4.5; 30 µg of PSMA-11 precursor, and 5 mg of ascorbic acid added as a scavenger. After 5 min of incubation at 40 °C, the reaction mixture is diluted with 5 mL of ppi water (Monico S.p.A. Via Ponte di Pietra, 7-30173 Venezia/Mestre). The solution is first passed through a terminal purification cartridge with cation exchange resin, which retains any free ^68^Ga^3+^, and then through a sterilizing filter of 0.22 µm (Millex-GV, Merck KGaA©, Darmstadt, Germany).

### 2.3. ^68^Ga-PSMA-11 PET/CT Scan Acquisition

^68^Ga-PSMA-11 PET/CT scans were performed on a Biograph mCT Flow^®^ machine (Siemens Healthineers, Erlangen, Germany). Acquisition was made on Flow mode (0.7 mm/sec) in 3D mode, with 3 mm slice thickness. Each patient was advised to have an empty rectum prior to PET/CT. Body-weighted ^68^Ga-PSMA-11 activity was intravenously administered (activity range: 100–200 MBq). Patients were required to drink 0.5 to 1 L of water during the uptake time to dilute the concentration of ^68^Ga-PSMA-11 in the urinary tract.

According to EANM guidelines, images started 60 ± 10 min after ^68^Ga-PSMA-11 injection [9].

PET/CT scan was performed with patients in the supine position on a carbon fiber flatbed, immobilized with a belly board device to allow for radiotherapy CT registration. A PET/CT scan was performed from the skull vertex to mid-thigh. A low-dose CT scan was performed for attenuation correction of the PET emission data.

In patients with a doubtful finding and/or significant bladder stagnation, a late acquisition was performed (about 120 ± 20 min after the tracer injection).

### 2.4. Image Analysis

All ^68^Ga-PSMA-11 PET/CT images were analyzed on a dedicated, commercially available software (Syngovia; Siemens Healthcare, Erlangen, Germany), which allowed for the review of PET, CT, and fused imaging data. Ga-PSMA focal uptake that was higher than adjacent background and not correlated with physiological tracer sites of uptake was categorized as suggestive of PCa recurrence. ^68^Ga-PSMA-11 uptake was considered physiological in lacrimal and salivary glands, liver, spleen, kidneys, and small intestine. PET/CT images were reviewed by two nuclear medicine physicians, who had at least 5 years of experience in reading PET/CT images and were unblinded to all available clinical data, and ^68^Ga-PSMA-11 data were reported according to procedure guidelines [9].

### 2.5. Radiotherapy Protocol

All patients were irradiated at the PET-positive locoregional recurrences. The GTV (gross tumor volume) was contoured according to ^68^Ga-PSMA-11 PET/CT pathologic volume. The planning target volume (PTV) was generated by adding a 3 mm isometric margin to the GTV. Patients were advised to have an empty rectum prior to receiving stereotactic body radiation therapy (SBRT) treatment, delivered by a tomotherapy unit. Daily image-guided radiation therapy (IGRT) was used. A median dose of 18 Gy in 3 consecutive daily fractions (IQR 18–21 Gy) was delivered. The prescription dose was calculated, taking into account the radiation dose previously received by the organs at risk (OAR; Figure 1).

### 2.6. Statistical Analysis

For the analysis of the response to radiotherapy, we considered biochemical recurrence-free survival (b-RFS), calculating the time interval from the last day of RT to the evidence of biochemical recurrence, defined as an increase in the PSA value ≥ 0.2 ng/mL above the PSA nadir after RT. Follow-up was performed according to institutional protocols, with regular serum PSA measurements and clinical follow-up visits. RT-associated toxicity was analyzed using the National Terminology Institute Common Terminology Criteria for Adverse Events (CTCAE) v4.0. For statistical analysis, Statdirect was used. We used the paired Student’s *t*-tests to compare the pre-RT and post-RT variables and the Wilcoxon signed-rank test when the data were not normally distributed. The time-to-event data were calculated using the Kaplan–Meier method and compared using the log-rank test.

## 3. Results

### 3.1. Overall Results

Thirty-eight patients (*n* = 38) were enrolled. The patients’ characteristics are described in detail in Table 1.

The overall median PSA level before SBRT was 1.10 ng/mL (IQR 0.82–2.59) and showed a significant decrease 3 and 6 months after treatment, with a median value of 0.60 ng/mL (IQR 0.31–0.96; *p* < 0.0001) and 0.51 ng/mL (IQR 0.29–1.17; *p* < 0.0001), respectively.

A nonparametric Friedman analysis showed no significant difference between the 3-month and 6-month postradiotherapy PSA values (*p* = 0.31).

### 3.2. Survival and Subpopulation Analysis

Median preradiotherapy PSA values resulted 1.12 ng/mL (IQR 0.70–2.82) in Subset 1 and 1.07 ng/ml (IQR 0.96–2.02) in Subset 2. Three months after SBRT, the median PSA value was, respectively, 0.49 ng/mL (IQR 0.31–0.95) in Subset 1 and 0.86 ng/mL (IQR 0.45–0.95) in Subset 2 (Figure 2).

In 20 out of 23 patients of Subgroup 1 (87%), the 3-month PSA value was reduced to >50% in 13 (65%), between 0–24.99% in 3 patients (15%), and between 25–50% in 4 patients (20%), whereas 3 patients showed PSA progression at the 3-month follow-up (increase in PSA of 52.1%, 8.7%, and 149%, respectively).

At 6 months, in 14/20 patients (70%), PSA reduction was confirmed, and, at the 9-month follow–up, 9/20 patients (45%) were in the biochemical disease control range.

Additionally, 13/15 (87%) patients belonging to Subgroup 2 had a PSA reduction 3 months after the end of radiotherapy (ΔPSA > 50% in 6 patients, variation between 25% and 50% in 5 patients, stable in 3 patients) while 2 patients experienced PSA progression. At 6 months, in 11/13 patients, PSA reduction was confirmed, and, at 9 months, 4 patients remained under biochemical disease control.

Raw data of the enrolled patients are reported in detail in Appendix A.

The overall biochemical recurrence-free survival (b-RFS) was 15.0 months (95% CI 13–23): there were no significant differences between the two subsets of patients considered (16 months, 95% CI 11–35 in Subset 1; 14 months, 95% CI 9-not reached in Subset 2 (*p* = 0.398; Figure 3).

All patients with PSA failure after SBRT (26/38) performed a further ^68^Ga-PSMA-11 PET/CT, which was able to find the disease location(s) in all cases. In particular, a persistence of the disease was documented in 4 patients, whereas in the other 22 patients, ^68^Ga-PSMA-11 PET/CT showed extrapelvic lymph nodes in 16 patients and bone lesions in 6 patients.

Only 5 patients with single lymph node relapse were further treated with SBRT, while the other patients were referred to ADT.

### 3.3. Toxicity

Briefly, 26 patients showed no toxicity, while 12 patients (31.6% of cases) developed grade 1 toxicity. The side effects highlighted were mainly fatigue in 3 patients, diarrhea in 6 patients, mild urinary incontinence and nicturia in 2 patients, which were resolved within 6 months of the end of therapy. In one patient, exacerbation of a previous hemorrhoidal syndrome occurred, with hematochezia resolved in a month. No grade 2 to 5 toxicity was observed.

## 4. Discussion

The management of BCR patients has been partially changed by the introduction of ^68^Ga-PSMA-11 PET/CT imaging, which allows for the early detection and targeting of small-sized PCa relapsing sites, including patients with PSA levels as low as 0.2 ng/mL. The simultaneous technological improvement of external beam radiotherapy techniques (EBRTs) has led to a growing interest in the use of these methods even in patients who have already been treated with radiation therapy, in particular by using dose hypofractionation to reduce toxicity. Accordingly, between June 2016 and December 2018, we evaluated consecutive postsalvage BCR patients with ^68^Ga-PSMA-11 PET/CT 76, with the aim of identifying disease recurrence amenable to be treated with a second local radiation cycle. Our data showed that retreating patients with SBRT guided by ^68^Ga-PSMA-11 PET/CT is a safe procedure, allowing us to adequately control BCR with an average time to further BCR of 15 months.

We did not observe significant differences in terms of b-RFS in patients with low-risk and intermediate-risk disease (respectively, ISUP 1–2 and ISUP 3).

Three months after SBRT completion, a statistically significant PSA decrease in both Subsets 1 and 2 was observed. G1 toxicity was reported in 13 patients only. Urinary acute toxicity in 3 patients and 7 rectal events were resolved in a few months from treatment completion.

To the best of our knowledge, this is the first study investigating the application of ^68^Ga-PSMA-11 PET/CT-guided SBRT retreatment in BCR patients who have already received sRT, presenting locoregional recurrence only.

Other studies evaluating the effectiveness of retreatments in BCR patients after radical treatment and S-RT have often been conducted in heterogeneous cohorts, including both local recurrences and skeletal or nodal metastases.

There are a few studies on reirradiation of intraprostatic recurrence in a limited group of patients. Jereczek-Fossa et al. reported a cohort of 34 patients (38 lesions) treated with CyberKnife-based SRT for isolated recurrent primary, lymph node, or metastatic prostate cancer. In this group, there were 15 patients with biopsy-proven intraprostatic relapse who were treated with CyberKnife: 9 patients were treated with CyberKnife-based SRT alone, while the other 6 patients had associated CyberKnife-based SRT + ADT; 6 out of the 9 patients revealed complete biochemical response and no rectal toxicity [10].

Oehus et al. recently published data from a multicenter retrospective study on 78 patients with biochemical progression after RP plus SRT and subsequent diagnosis of oligo-recurrent PCa assessed by PSMA-based PET imaging. Although the radiation therapy scheme was variable, a significant reduction in PSA levels (1.90 ng/mL versus 0.88 ng/mL, *p* = 0.008) was observed. This study, however, included patients with oligo-metastatic disease in both the prostate bed and at other sites (skeleton, viscera), and a number of patients (17%) received concomitant antiandrogen therapy [11]. On the other hand, international guidelines suggest the use of ADT in metastatic patients, which, however, is not exempt from well-known side effects [12].

The advantage in terms of the effectiveness of hormone-based therapy and radiation therapy alone, compared to the association of the two, need to be clarified. Berkovic et al. in 2013 issued one of the first studies reporting on the interesting end-point of deferring systemic treatment. Three synchronous asymptomatic metastases were diagnosed in 24 BCR patients after treatment with curative intent (RP, S-RT, or a combination of both). All patients completed the SBRT protocol with a median dose of 50 Gy and a median follow-up of 24 months. ADT was initiated if more than 3 metastases were detected during the follow-up, resulting in an ADT-FS of 82% at 1 year and 54% at 2 years [13]. Recently, Ost et al. reported significantly longer androgen deprivation therapy (ADT)-free survival in a cohort of 62 oligo-recurrent PCa patients who underwent radiation therapy on locoregional recurrence compared to surveillance. The median ADT-free survival period was 13 months for the surveillance group vs. 21 months for the locoregional therapy group. In total, 74% of patients treated with locoregional therapy had a PSA decline compared with 42% in the surveillance arm [14]. Other studies, using choline PET for randomization, have shown that local treatment alone could significantly delay the start of androgen therapy [15,16]. Data emerging from the Stampede study suggest that concomitant use of both ADT and RT could improve b-RFS and overall survival (OS) only for patients with low-volume PCa recurrence, while in patients with higher metastatic burdens, RT did not improve OS [17].

Consequently, therapeutic approaches meant to defer ADT-induced morbidity should be considered in patients with low-volume metastatic disease. In this regard, our patients were restaged with ^68^Ga-PSMA-11 PET/CT, allowing us to identify a low-metastatic burden in 43 out of 76 (57%) BRCs tested; 17/76 (22%) had ^68^Ga-PSMA-11 PET/CT positivity outside the locoregional bed, and 16/76 patients were negative on ^68^Ga-PSMA-11 PET/CT (21%). In the present study, we focused our attention on the 43 patients with low tumor volume and investigated the activity of r-RT alone in this subset of cases. The results obtained increased the impression that hormone therapy could be delayed when low tumor burden is documented. For those patients with oligo-metastatic cancer, who are unfit for RT + ADT, an alternative therapeutic plan should be envisaged.

### Limitation

This study is not exempt from limitations. One of the main limitations is represented by the retrospective design of this analysis and by the limited sample size. In addition, all patients undergoing r-RT had only a prostate bed positive PET scan; patients with negative PET scans were not considered in the analysis.

## 5. Conclusions

Our results confirm the role of ^68^Ga-PSMA-11-PET/CT as a diagnostic procedure able to identify the site of recurrence in patients who have failed S-RT in the prostate bed. ^68^Ga-PSMA-11 PET/CT-guided SBRT was effective and without major side effects in the majority of patients studied. Biochemical recurrence-free survival (b-RFS) was 15.0 months, and this delayed the initiation of ADT in patients who already received radiation treatments on the same site, thus supporting the use of metastases-directed radiotherapy as a safe and effective treatment.

## Figures and Tables

**Figure 1 biomedicines-08-00536-f001:**
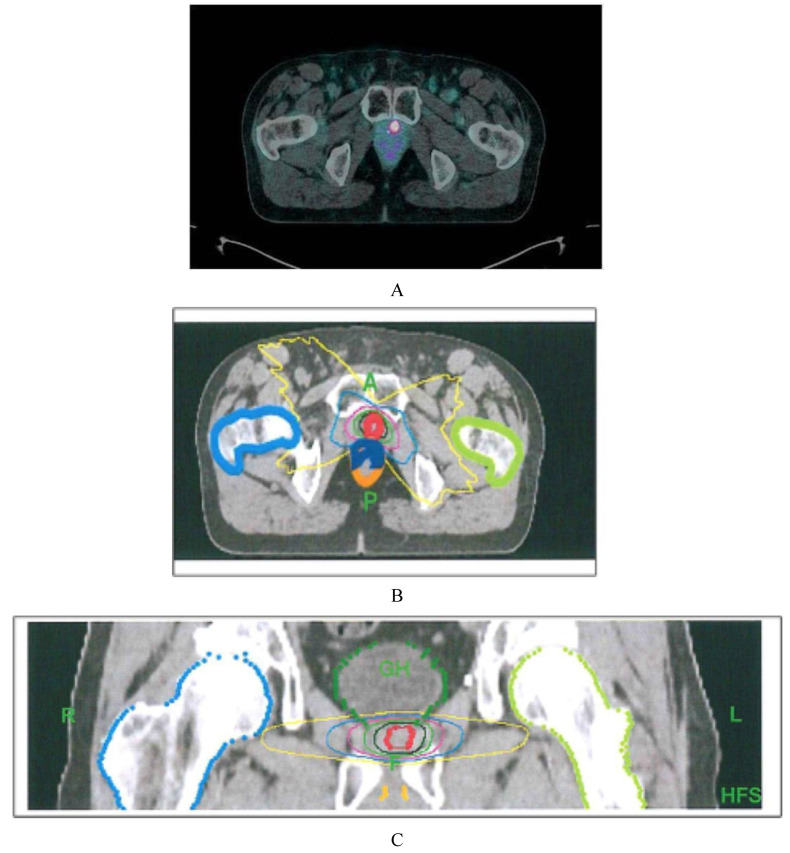
Patient characteristics: 79 yo, radical prostatectomy (RP) + radiotherapy (RT) in 2017, PSA increase 0.87 ng/mL (July 2018). (**A**) local relapse in left prostatic bed in prostate-specific membrane antigen-based positron emission computed tomography hybrid imaging (^68^Ga-PSMA-11 PET/CT). (**B**–**D**) GTV and PTV contoured according to ^68^Ga-PSMA-11 PET/CT in axial, coronal, and sagittal planes. (A: anterior; P: posterior; R: right: L: left; GH: gantry head; F: feet; HFS: head first supine). Anatomic structures: blue line: rectum anterior wall, orange line: rectum posterior wall; light blue line: right femur; green line: left femur. Cumulative isodose lines: red 18 Gy, dark green 15 Gy, light green 12 Gy, pink 9 Gy, blue 6 Gy and yellow 3 Gy.

**Figure 2 biomedicines-08-00536-f002:**
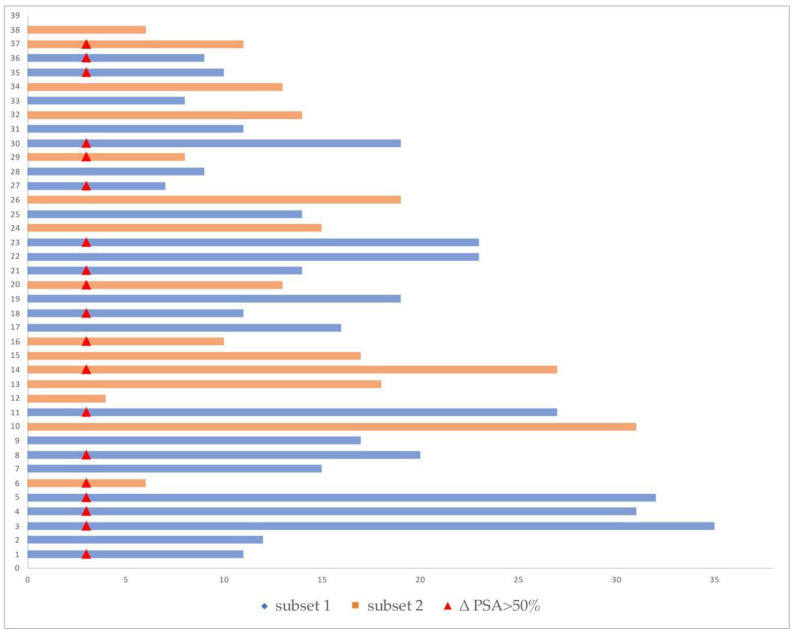
Swimmers plot with time to new biochemical recurrence (BCR) in Subset 1 (blue line) and Subset 2 (orange line) after r-RT. The red points represent patients with a PSA reduction of more than 50% three months after r-RT.

**Figure 3 biomedicines-08-00536-f003:**
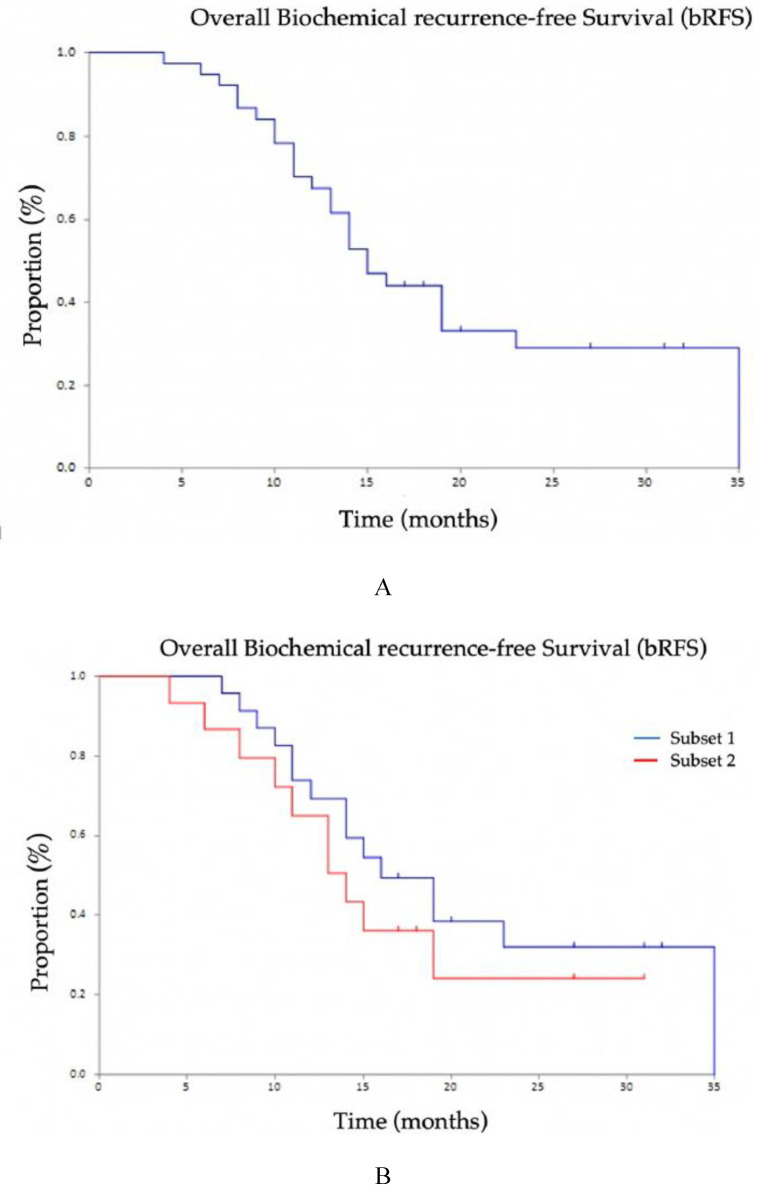
Kaplan–Meier curves of biochemical recurrence-free survival (b-RFS) after ^68^Ga-labeled PSMA ligand PET-directed radiotherapy of prostate cancer: (**A**) overall population; (**B**) Subset 1 and Subset 2.

**Table 1 biomedicines-08-00536-t001:** Patients characteristics.

Variable	Overall *n* = 38 Median (IQR)
Age (years) at Ga-PSMA PET/CT	75 (71–80 years)
Pre RT PSA	1.10 ng/mL (0.82–2.59 ng/mL)
3 months post-RT PSA	0.60 ng/mL (0.31–0.96 ng/mL)
6 months post-RT PSA	0.51 ng/m (0.29–1.17 ng/mL)
Primary therapies	*n* (%)
RT	11 (28.9%)
RP + RT	27 (71.1%)
ISUP grade	
1	10 (26.3%)
2	13 (34.2%)
3	15 (39.5%)
ISUP subset	
1 (ISUP grades 1 and 2)	23 (60.5%)
2 (ISUP grade 3)	15 (39.5%)
T status	
2	16 (42.1%)
3	22 (57.9%)
N status	
0	22 (57.9%)
1	8 (21.05%)
x	8 (21.05%)

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
