# Peer review of "^68^Ga-PSMA-11 PET/CT-Guided Stereotactic Body Radiation Therapy Retreatment in Prostate Cancer Patients with PSA Failure after Salvage Radiotherapy"

_biomedicines, 2020, doi:10.3390/biomedicines8120536_

Round 1
Reviewer 1 Report
The authors describe the use of 68Ga-PSMA-11 PET/CT for radiotherapy retreatment in PC patients with PSA failure after salvage radiotherapy. Nevertheless, the setup/methods as well as presentation are in my opinion of good quality there are some minor flaws which should be corrected before publication.
Please use the reference style requested by the journal.
Please follow the nomenclature guidelines for radiopharmaceuticals of the srs. (https://static1.squarespace.com/static/59bd4d82d7bdce156a52b6bd/t/59bd51fd3bb053f5fd2f2c89/1494418606877/9May2017Consensus-nomenclature-rules.pdf)
Please revise for missing or additional spaces (e.g. Line 59: 1.0ng/ml).
Please revise for formatting errors (e.g. Line 99: 68Ge/68Ga generator; Line 113: flatbed).
Please be consistent in writing units (e.g. Line 141 0,2 ng / ml; Line 154 0.50 ng/mL 0.44 ng/ml)
Abstract: The word limit according to the instructions for authors is 200. Please shorten the abstract (>300 words).
Materials & Methods: 2.2 Please elaborate the production of the tracer adequately or reference to literature where it is described completely. What was the amount of PSMA-11 used? Which SCX resin was used? Which post-processing was used? What amount of which buffer was used? When it was a commercial available method/process including consumables and cassettes please provide the name of the provider thereof (Eckert&Ziegler? abx?). Which temperature was used? Normally, the radiolabelled peptides are purified using reverse phase chromatography instead of cation-exchange. I guess instead of the cation-exchange cartridge a C18 SPE cartridge (waters?) was applied. Please revise. What is meant with an appropriate aseptic formulation (dilution in isotonic saline? with/without radiolysis inhibitor? followed by sterile filtration?).
2.3 Line 111 70 min uptake time?
2.4. Sentence 2 (Line 118 to 120) and last sentence (Line 123 to 125) are redundant?
Author Response
The authors describe the use of 68Ga-PSMA-11 PET/CT for radiotherapy retreatment in PC patients with PSA failure after salvage radiotherapy. Nevertheless, the setup/methods as well as presentation are in my opinion of good quality there are some minor flaws which should be corrected before publication.
Please use the reference style requested by the journal. : Done
Please follow the nomenclature guidelines for radiopharmaceuticals of the srs. (https://static1.squarespace.com/static/59bd4d82d7bdce156a52b6bd/t/59bd51fd3bb053f5fd2f2c89/1494418606877/9May2017Consensus-nomenclature-rules.pdf) Done
Please revise for missing or additional spaces (e.g. Line 59: 1.0ng/ml). Done
Please revise for formatting errors (e.g. Line 99: 68Ge/68Ga generator; Line 113: flatbed). Done
Please be consistent in writing units (e.g. Line 141 0,2 ng / ml; Line 154 0.50 ng/mL 0.44 ng/ml) Done
Abstract: The word limit according to the instructions for authors is 200. Please shorten the abstract (>300 words). as requested, we have drafted the abstract respecting the allowed word limit.
Materials & Methods: 2.2 Please elaborate the production of the tracer adequately or reference to literature where it is described completely. What was the amount of PSMA-11 used? Which SCX resin was used? Which post-processing was used? What amount of which buffer was used? When it was a commercial available method/process including consumables and cassettes please provide the name of the provider thereof (Eckert&Ziegler? abx?). Which temperature was used? Normally, the radiolabelled peptides are purified using reverse phase chromatography instead of cation-exchange. I guess instead of the cation-exchange cartridge a C18 SPE cartridge (waters?) was applied. Please revise. What is meant with an appropriate aseptic formulation (dilution in isotonic saline? with/without radiolysis inhibitor? followed by sterile filtration?). As requested, we have proceeded to modify the part relating to the synthesis of the radiopharmaceutical in the text
2.3 Line 111 70 min uptake time?
PET scan was performed 60 +/- 10 minutes after 68Ga-PSMA administration according EANM guidelines.
Wolfgang P. Fendler, Matthias Eiber, Mohsen Beheshti, Jamshed Bomanji, Francesco Ceci, Steven Cho, Frederik Giesel, Uwe Haberkorn, Thomas A. Hope, Klaus Kopka, Bernd J. Krause, Felix M. Mottaghy, Heiko Schöder, John Sunderland, Simon Wan, Hans-Jürgen Wester,Stefano Fanti,,Ken Herrmann 68Ga-PSMA PET/CT: Joint EANM and SNMMI procedure guideline for prostate cancer imaging: version 1.0 Eur J Nucl Med Mol Imaging (2017) 44:1014–1024
2.4. Sentence 2 (Line 118 to 120) and last sentence (Line 123 to 125) are redundant? Done
Reviewer 2 Report
The emergence of next-generation imaging techniques such as PSMA PET which can more accurately identify oligometastatic disease, together with treatment technologies more capable of delivering metastasis-directed therapies with low toxicities, represent an inflection point in developments on prostate cancer management. However, firm data on how PSMA PET will affect the oncological outcomes of oligometastatic patients after radical treatment is missing. Caroli and colleagues retrospectively investigated the impact of 68Ga-PSMA-11 PET/CT on treatment decision-making and biochemical recurrence-free survival in prostate cancer patients affected by PSA failure after salvage-radiotherapy, in biochemical recurrence setting. They enrolled 43 consecutive patients with pathological PSMA uptake limited within the prostate bed/prostate and subsequently treated with stereotactic radiotherapy (SBRT). Overall, a significant PSA decrease was found after treatment at 3 and 6 months. Biochemical recurrence-free survival was 15.0 months (95% CI: 12.19 – 17.80) with a median follow-up time of 14.0 months (range 4 – 35).
The study is well designed and clearly presented, reporting data of interest for the uro-oncology community. Nevertheless, some issues give rise to concern and should be clarified.
Major
- the first aim of the study is “to investigate the impact of 68Ga-PSMA-11 PET/CT on treatment decision-making”. How the authors conducted this analysis is however not clear alongside the whole manuscript, and the Materials and Methods section has to be implemented accordingly. Otherwise, this aim should be deleted in the abstract and introduction. The authors take into account only positive PSMA PET scans, and with prostate bed confined PSMA positive disease. All patients enrolled underwent SBRT. This methodology limits the analysis on PSMA-based treatment decision-making. A cohort not treated, or treated with a different approach, or imaged with another technique is missing. Did the patients have a positive MRI (or US) on the prostate bed/prostate? More data are needed, otherwise move as a secondary aim or delete.
- A major limit of 68Ga-PSMA-11 PET/CT is the prostate bed exploration due to the urinary excretion of the tracer. Which criteria were employed to assess the prostate bed? Is available the rate of agreement between the two readers? Was adopted a dedicated acquisition PET protocol to avoid/diminish the presence of radioactive urine in addition to the hydration reported (from 0.5 to 1L)?
- All patients were re-treated after salvage-radiotherapy and some patients underwent also primary EBRT. More details on the Material and Methods section should be given, in order to explain how critical organs were approached, i.e. urethra, bladder and rectum. Did you have a sum plan that consider also previous radiation therapies for these three organs? Only after detailing the radiotherapeutic approach, data on toxicity can be accurately interpreted;
- Interestingly, the median follow-up was 14.0 months (range 4 – 35) and the biochemical recurrence-free survival resulted 15.0 months, longer than the median value. Please give an explanation.
Minor
- Title: add that patients were re-treated with SBRT in order to avoid misinterpretation;
- Material and Methods: “All patients had previously undergone curative prostatic radiotherapy or…” this means that some patients underwent primary EBRT, salvage-RT and SBRT to the prostate. If yes, please give more information and clearly explain because three radiation-based treatment on the prostate are not so common and completely safe;
- Values are presented with median, thus consider using interquartile range (IQR) instead of range;
- Discussion: specify that the STOMP trial employed choline PET;
- Discussion: rows 265-271 is a bit inappropriate, the first two/three sentences are more adequate for the introduction, they refer to what we already know. The last sentence should be used as a prospective at the end of the discussion, even if a bit futuristic;
- Table 1: please add percent (%) values to the ISUP grade column;
- Please adopt the same word/acronym for salvage radiotherapy and another one for stereotactic radiotherapy to reduce the risk of misinterpretation. Alongside the manuscript different word are used, i.e. s-RT (abstract), S-RT (abstract), radiotherapy (abstract), SBRT (m&m), EBRT (results);
- Please review the whole manuscript, some minor grammatic mistakes need to be correct, e.g. 68Ga, pts instead of patients (row 254); and some percent (%) have to be added, e.g. row 173 “In 20 out of 23 patients…” give (%).
Author Response
Major
Q: the first aim of the study is “to investigate the impact of 68Ga-PSMA-11 PET/CT on treatment decision-making”. How the authors conducted this analysis is however not clear alongside the whole manuscript, and the Materials and Methods section has to be implemented accordingly. Otherwise, this aim should be deleted in the abstract and introduction. The authors take into account only positive PSMA PET scans, and with prostate bed confined PSMA positive disease. All patients enrolled underwent SBRT. This methodology limits the analysis on PSMA-based treatment decision-making. A cohort not treated, or treated with a different approach, or imaged with another technique is missing. Did the patients have a positive MRI (or US) on the prostate bed/prostate? More data are needed, otherwise move as a secondary aim or delete.
A: thanks for the observation. We agree with you: the population considered is extremely selected and therefore probably the statement about the ability of PSMA PET to modify the management of patients appears not completely supported by our data.
Therefore, in relation to this evidence, we made a change in the declination of the primary objective of the study.
Q: A major limit of 68Ga-PSMA-11 PET/CT is the prostate bed exploration due to the urinary excretion of the tracer. Which criteria were employed to assess the prostate bed? Is available the rate of agreement between the two readers? Was adopted a dedicated acquisition PET protocol to avoid/diminish the presence of radioactive urine in addition to the hydration reported (from 0.5 to 1L)?
A: Thanks for the suggestions. We have provided the required information in the text in the materials and methods section. The urinary excretion of the tracer can certainly limit the correct visualization of the prostate loggia, often causing a reduction in the sensitivity of the method. In order to improve the accuracy of the method in patients with a doubtful finding and with significant bladder stagnation, a late acquisition was performed (about 120 after the injection).
Q: All patients were re-treated after salvage-radiotherapy and some patients underwent also primary EBRT. More details on the Material and Methods section should be given, in order to explain how critical organs were approached, i.e. urethra, bladder and rectum. Did you have a sum plan that consider also previous radiation therapies for these three organs? Only after detailing the radiotherapeutic approach, data on toxicity can be accurately interpreted;
A: We didn't get a sum plan. The goal was to minimize the dose to the rectum, bladder and small intestine.
Dose-volume objectives required during plan optimization for organs at risk (OARs) were: Rectum: D max = 12 Gy and V6 Gy<5%
Bladder: Dmax = 15Gy and V10Gy<5%
Small intestine: V18 <2 cm3
Q: Interestingly, the median follow-up was 14.0 months (range 4 – 35) and the biochemical recurrence-free survival resulted 15.0 months, longer than the median value. Please give an explanation.
A: You're right, sorry for the transcription error. In fact, the median follow-up is 27 months (range 4-35): we therefore proceeded to modify the text.
Minor
- Title: add that patients were re-treated with SBRT in order to avoid misinterpretation;
- Done
- Material and Methods: “All patients had previously undergone curative prostatic radiotherapy or…” this means that some patients underwent primary EBRT, salvage-RT and SBRT to the prostate. If yes, please give more information and clearly explain because three radiation-based treatment on the prostate are not so common and completely safe;
- You are right: the sentence as it was written can generate the doubt that all patients may have undergone three different courses of radiotherapy. In reality, there are two possible conditions: patients undergoing EBRT or subjects undergoing prostatectomy and sRT. We have therefore rephrased the sentence.
- Values are presented with median, thus consider using interquartile range (IQR) instead of range;
- Done
- Discussion: specify that the STOMP trial employed choline PET
- Done
- Discussion: rows 265-271 is a bit inappropriate, the first two/three sentences are more adequate for the introduction, they refer to what we already know. The last sentence should be used as a prospective at the end of the discussion, even if a bit futuristic;
- We agree with you that the phrase may be futuristic, even if the association between radiation therapy and metabolic radiotherapy could represent an interesting topic to be explored. For this reason we preferred to delete the entire sentence.
- Table 1: please add percent (%) values to the ISUP grade column;
- Done
- Please adopt the same word/acronym for salvage radiotherapy and another one for stereotactic radiotherapy to reduce the risk of misinterpretation. Alongside the manuscript different word are used, i.e. s-RT (abstract), S-RT (abstract), radiotherapy (abstract), SBRT (m&m), EBRT (results);
- Done
- Please review the whole manuscript, some minor grammatic mistakes need to be correct, e.g. 68Ga, pts instead of patients (row 254); and some percent (%) have to be added, e.g. row 173 “In 20 out of 23 patients…” give (%).
- Done
Reviewer 3 Report
The present manuscript aims to assess the role of ga68-PSMA for the identification of the only recurrence in the prostatic bed and the impact of this results on the progression free survival (in terms of PSA) in patients with BCR of prostate cancer, after primary treatments.
Some comments for the improvement of the paper:
1) the inclusion criteria and the exclusion criteria are not well reported, because the authors identified only patients with a positive 68Ga-PSMA PET/CT in the prostatic bed, without the evidence of distant or loco-regional lymph node or organ metastases, but after in the results they excluded patients with ISUP 3.
2) information about acquisition protocol are missing (early or late acquisition PET were performed?) and also the interpretation (the presence of tracer in the bladder can render difficult the identification of local recurrence)
3) some sentences have been reported twice (5-year experience and also the final number of enrolled patients in the results paragraph)
4) the final population should be composed by 38 patients rather than 43.
5) a comparison, in terms of PSA recurrence between positive 68Ga-PSMA in fossa and negative 68Ga-PSMA PET would be very useful, in order to better understand the real negative predictive value of the imaging for the identification of distant metastases.
6) Did the imaging change the patient management in these patients? or the patients would be treated the same with rRT, independent from the results of PSMA? some comments are necessary, mainly by considering the population with a negative PSMA PET.
Author Response
1) the inclusion criteria and the exclusion criteria are not well reported, because the authors identified only patients with a positive 68Ga-PSMA PET/CT in the prostatic bed, without the evidence of distant or loco-regional lymph node or organ metastases, but after in the results they excluded patients with ISUP 3.
- Thanks for the suggestions. We have provided the required information about inclusion criteria and the exclusion criteria in the text in the materials and methods section. In the results we decided to exclude ISUP 3 patients from the final analysis because, as the guidelines indicate, they also started hormone therapy.
2) information about acquisition protocol are missing (early or late acquisition PET were performed?) and also the interpretation (the presence of tracer in the bladder can render difficult the identification of local recurrence).
- Thanks for the suggestions. We have provided the required information in the text in the materials and methods section. The urinary excretion of the tracer can certainly limit the correct visualization of the prostate loggia, often causing a reduction in the sensitivity of the method. In order to improve the accuracy of the method in patients with a doubtful finding and with significant bladder stagnation, a late acquisition was performed (about 120 after the injection).
3) some sentences have been reported twice (5-year experience and also the final number of enrolled patients in the results paragraph).
- Done
4) the final population should be composed by 38 patients rather than 43.
- Thanks for the suggestions. We proceeded with a partial modification of the data of the series because the actual number of patients analyzed was unclear
5) a comparison, in terms of PSA recurrence between positive 68Ga-PSMA in fossa and negative 68Ga-PSMA PET would be very useful, in order to better understand the real negative predictive value of the imaging for the identification of distant metastases.
- Thanks for the suggestion. We agree with you that whereas there is accumulating evidence on the clinical impact of a positive 68Ga-PSMA PET / CT in guiding the treatment strategy in BCR patients, little is known on the predictive value of a negative 68Ga-PSMA PET / CT scan in biochemically relapsing patients. Recently our group published a work on the negative predictive role of a 68Ga-PSMA PET / CT in the management of biochemical recurrent prostate cancer patients (Celli et Al. EJNMMI 2020 doi.org/10.1007/s00259-020-04914-8) . In this study, based on 103 patients, with a median overall level of PSA at the time of 68Ga-PSMA PET / CT scan equal to 0.47ng / ml [range 0.196-5.2 ng / ml], we showed how the only significant risk factor for clinical relapse was the primary PCa ISUP grade. Patients with indolent primary PCa biology (ISUP grades 1 and 2) benefited from a significantly longer clinical relapse-free survival (CRFS) compared to patients with intermediate (ISUP grade 3) and aggressive PCa (ISUP grades 4 and 5). In our study, even if the PSA level did not significantly correlate with CRFS, patients with PSA higher than 1.0ng / ml experienced a poorer 24-month CRFS compared to patients with PSA levels between 0.5 and 1.0ng / ml (24-month CRFS was 22.5% and 45.6%, respectively). Surely a prospective study comparing the two patient series can be useful for a better definition of the NPV of Ga-PSMA PET/CT.
6) Did the imaging change the patient management in these patients? or the patients would be treated the same with rRT, independent from the results of PSMA? some comments are necessary, mainly by considering the population with a negative PSMA PET.
- Thanks for the suggestions. Currently there is no consensus regarding the optimal management for locally recurrent PCa. The most often considered strategy is androgen deprivation therapy (ADT) with negative impact on the
quality of life and tumour control of limited duration.
Local salvage approaches with curative intent, such as salvage
prostatectomy, re-irradiation, cryotherapy and high intensity focused
ultrasound (HIFU) have shown to be associated with a high rate of
severe side effects for normal tissue complication mostly after high
radiation dose and elderly age.
Stereotactic body radiation therapy (SBRT) has emerged as a potential
therapeutic option for limited/locally recurrent prostatic cancer due
to its dosimetric and radiobiologic advantages. In order to limit
normal tissue toxicity, the target volume must be strictly limited to
the recurrent macroscopic tumor. The information obtained by the
PSMA CT / PET study is crucial in identifying the area of recurrence
and treating it in a focal way.
Round 2
Reviewer 2 Report
I would really thank you authors for their efforts to improve on the manuscript quality. Very well done.
No further comments from my side.
Reviewer 3 Report
The manuscript is now improved and it can be accepted in the present form.